# Mechanism of Oxytetracycline Removal by Coconut Shell Biochar Loaded with Nano-Zero-Valent Iron

**DOI:** 10.3390/ijerph182413107

**Published:** 2021-12-12

**Authors:** Qi Li, Siyu Zhao, Yuhang Wang

**Affiliations:** College of Urban and Environmental Sciences, Northwest University, Xi’an 710127, China; 201931906@stumail.nwu.edu.cn (S.Z.); yuhangwang895@stumail.nwu.edu.cn (Y.W.)

**Keywords:** biochar, oxytetracycline, nano-zero-valent iron, adsorption, degradation

## Abstract

In this paper, coconut shell biochar (BC), pickling biochar (HBC), and nano-zero-valent iron-loaded biochar (nZVI-HBC) were prepared; these were used to remove oxytetracycline (OTC), and the removal mechanism and degradation product were analyzed. These biochars were characterized using SEM, XRD, FTIR, and XPS. The effects of biochar addition amount, pH, ion type, and ion concentration on OTC adsorption were studied by a batch adsorption experiment. Under the optimal conditions, the equilibrium adsorption capacity of nZVI-HBC to OTC was 196.70 mg·g^−1^. The adsorption process can be described by Langmuir isothermal adsorption equations, conforming to the pseudo-second-order dynamics model, indicating that adsorption is dominated by single-molecule chemical adsorption, and a spontaneous process of increasing heat absorption entropy. Mass spectrometry showed that the OTC removal process of nZVI-HBC included not only adsorption but also degradation. These results provide a practical and potentially valuable material for the removal of OTC.

## 1. Introduction

In recent years, with the rapid development of aquaculture, the pollution of antibiotics in the water environment has become increasingly more serious [1,2]. Due to the complex molecular structure and numerous intermediate products of antibiotics, they can be harmful to biological health [3,4]. Among them, oxytetracycline (OTC) is one of the most widely used veterinary antibiotics; aquaculture wastewater constantly enters the environment and gradually accumulates, where long-term exposure can disrupt the balance of the ecosystems, leading to pathogenic microorganisms developing resistance [5,6]. Some research data show that OTC excreted through feces and urine remains bioactive, and over 90% is not metabolized. Today, more than 70% of OTC products end up in the environment after treatment at wastewater treatment plants [7]. According to the survey, OTC is widely used in veterinary medicine and food production, accounting for 70% of the consumption of antibiotics in Europe in 2017. Moreover, the OTC concentration detected in Chinese aquaculture was 315–15,163 ng·L^−1^ [8,9]. However, the traditional water pollution removal methods have no noticeable removal effect on antibiotics such as OTC. Therefore, it is necessary to find a suitable way to remove aquaculture wastewater.

At present, there are several methods for the removal of antibiotics, such as adsorption [10,11], biological treatment [12], biodegradation [13], membrane separation [14], and oxidation [15]. Zhou et al. reported that the MnO_2_/UIO-66 composites prepared through an advanced oxidation process have great potential for application in the degradation removal of OTC [16]. Hadki et al. used the reduction of boron oxide graphene (B-rGO) to remove OTC, and in the first ten minutes, the removal rate could reach more than 85% [17]. Lian et al. used FeO_n_(OH)_m_-modified oyster shell powder for OTC conversion. The effect was more than 81.5% [18]. Jia et al. used the coconut shell biochar adsorption of OTC in aqueous solutions, and the results showed that the maximum adsorption rate of OTC reached 1667 mg·kg^−1^ [19]. Among them, the adsorption method has widely been used because of its advantages of being a simple process, no secondary pollution, a comprehensive source of adsorbent, and so on. Biochar, as a new adsorbent, is characterized by large specific surface area, rich surface functional groups, and mineral compounds, making it widely used in the removal of antibiotics in the water environments. Currently, some domestic and foreign scholars have used biochar to research the removal and degradation of antibiotics in the natural environment [20]. Compared with other biochar raw materials, coconut shell has the largest surface area and porosity. Meanwhile, the coconut shell has the advantages of low ash content, high density, and high mechanical strength, making it suitable as a raw material for the adsorption of pollutants [21,22]. Shen et al. reported that coconut shell biochar has a good removal effect on Cr (VI) due to its rich functional groups [23]. Moreover, coconut shell biochar as an improver for fixing heavy metals in contaminated soil can improve the physical and chemical properties of soils [24]. Coconut shell biochar and quartz sand as composite adsorbents can adsorb Mn with a high removal efficiency of 94.22% [25]. Therefore, coconut shell has many advantages, and it is selected as the raw material of biochar.

Nano-zero-valent iron (nZVI) refers to zero-valent iron particles with a particle size of 1–100 nm, which are characterized by small particle size, strong reducibility, large sur-face area, and strong transferability, etc. Thus, nZVI has good application prospects in the removal of water pollutants [26]. The research shows that nZVI has a good removal effect on antibiotics [27], heavy metals [28], and inorganic salts. However, nZVI has some limitations because of its characteristics, and the modification of nZVI is also one of the standard methods used by many researchers [29]. Among them, the carrier load [30,31] is one of the simple methods for the modification of nZVI. The carrier generally selects materials with abundant sources and a low price of chromium, which ensures the processing effect and controls the cost. It also has good prospects for application in industrial production [32].

In this paper, coconut shell was used as a raw material of biochar, coconut shell biochar (BC) was prepared, it was modified by hydrochloric acid (HBC), and nZVI was loaded on HBC by the liquid-phase reduction method (nZVI-HBC). At the same time, the OTC was removed in the batch adsorption experiment, and its adsorption properties of different materials were compared. Combined with material characterization, the adsorption mechanism of pollutants by different modification methods was explored, and the possible mechanism and other intermediates in the degradation process were analyzed. nZVI was loaded on the basis of modified biochar, which provided an efficient and economic treatment method for OTC removal and a reference for aquaculture wastewater.

## 2. Materials and Methods

### 2.1. Materials and Reagent

The coconut shell was collected from a planting base in Hainan, China. OTC (analytical grade, purity > 99%) was from Shanghai Source Biological Technology Co. Ltd., Shanghai, China. All reagents in this work were of analytical-grade purity and above and were bought from Comiou Chemical Reagent Co. Ltd., Tianjin, China. All ultrapure water used in the experiments had a resistivity of 18.2 MΩ.

### 2.2. Analytical Instruments and Methods

The surface structure and morphology of the samples were analyzed using a scanning electron microscope (FEI Quenta 400 FEG, Hillsboro, OR, USA). An X-ray diffraction analyzer (Rigaku Corporation, Rigaku MiniFlex Ⅱ, Tokyo, Japan) was applied to investigate the crystalline structure of biochar. The chemical properties of biochar were highlighted by a Fourier-transform infrared spectrometer (Nicolet Instruments, Nexus870, Madison, WI, USA). X-ray photoelectron spectroscopy (Rigaku Corporation, Rigaku MiniFlex II, Tokyo, Japan) was used to determine the elemental quantification and valence analysis in ferrocarbon.

The content of OTC was determined by an ultraviolet spectrophotometer (Shimadzu Corporation, UV-1800, Tokyo, Japan) with a maximum absorption wavelength of 268 nm [33]. The degradation products of OTC were determined by LC-MS (Agilent Technologies, Agilent 6460C, Santa Clara, CA, USA).

### 2.3. Experimental Method

#### Material Preparation

The preparation of the BC: The coconut shells were placed in the crucible and placed in the muffle furnace to heat at a rate of 5 °C·min^−1^ to 800 °C for 2 h. After the thermolysis, it was cooled to room temperature, and after passing 100 mesh sieves (<0.150 mm) in a sealed bag, it was recorded as BC.

The preparation of the HBC: BC was soaked in 1 mol·L^−1^ of HCl for 24 h, repeatedly cleaned to neutral with deionized water, dried for 24 h in a drying oven at 80 °C, and stored in a sealed bag, denoted as HBC.

The preparation of the nZVI-HBC: First, 0.56 g of HBC was placed in a three-neck flask, 100 mL of FeSO_4_·7H_2_O ethanol-water solution (ethanol: water = 3:7) was added, 2 mL of polyethylene glycol solution was dropped, and it was stirred for 30 min. Then, under the existence of nitrogen, 0.5 mol of NaBH_4_ was dropped into the solution and stirred violently. After the reaction was completed, stirring continued for 1 h to produce modified biochar load nZVI (nZVI-HBC) [34]. nZVI-HBC was adsorbed with magnets and washed with deoxygenated high-purity water and anhydrous ethanol 3 times. It was placed in a the vacuum drying oven at 60 °C for 24 h, and then stored in a brown bottle for backup.

### 2.4. Batch Adsorption Experiment

Experiment 1: Effect of biochar addition amount on OTC removal rate. Biochar was added to an OTC solution with an initial mass concentration of 20 mg·L^−1^ and a volume of 50 mL in the proportion of 2, 4, 6, 8, 10, 15, and 20 mg. A conical flask was placed in a constant-temperature oscillating chamber, and then it oscillated at a frequency of 150 r·min^−1^ at (25 ± 1) °C for 24 h in the dark. The liquid was taken after filtering by a 0.22 μm membrane sample, the concentration of OTC solution in the remaining solution was determined using ultraviolet spectrometry, and the average was repeated three times.

Experiment 2: Effect of different initial pH values on OTC removal rate. Here, 6 mg of biochar was placed in a conical flask, and added with 20 mg·L^−1^ and a volume of 50 mL of OTC solution was added. The pH was adjusted to 3.0, 5.0, 7.0, 9.0, and 11.0 using HCl or NaOH. The other settings were the same as experiment 1, and the average was repeated three times.

Experiment 3: Effect of ion type on OTC removal rate. Keeping the other conditions unchanged, 0.2 mol·L^−1^ of NaCl, KCl, MgCl_2_, CaCl_2_, NaNO_3_, NaHCO_3_, and Na_2_CO_3_ solution were prepared. The other settings were the same as experiment 1, and the average was repeated three times.

Experiment 4: Effect of ion concentration on OTC removal rate. Keeping the other conditions unchanged, the ionic strength of solution was adjusted by different concentrations of NaCl solution (0.00, 0.05, 0.10, 0.20, and 0.50 mol·L^−1^). The other settings were the same as experiment 1, which was repeated three times to average.

### 2.5. Isothermal Adsorption

The initial OTC solution concentrations were set as 5, 10, 20, 30, 40, and 50 mg·L^−1^, and biochar was added to different concentrations of OTC solutions. The conical flasks were placed in a constant-temperature shock chamber at 15 °C, 25 °C, and 35 °C, and then shaken at a frequency of 150 r·min^−1^ for 24 h. The liquid was then taken after filtering by a 0.22 μm membrane sample, and the samples were measured. The concentration of OTC solution in the remaining solution was determined by ultraviolet spectrometry. The experiment was repeated three times, and the average value was taken. The experimental results were fitted by the Langmuir isothermal adsorption model, Freundlich isothermal adsorption model, and Temkin isothermal adsorption model [35].
$$\mathrm{Langmuir}\;\mathrm{isothermal}\;\mathrm{adsorption}\;\mathrm{model}:\;q_{e}=\frac{q_{m}K_{L}C_{e}}{1+K_{L}C_{e}}$$
Freundlich isothermal adsorption model: qe=KFCe1n
$$\mathrm{Temkin}\;\mathrm{isothermal}\;\mathrm{adsorption}\;\mathrm{model}:\;q_{e}=\left(\frac{RT}{b}\right)\ln K_{T}+\left(\frac{RT}{b}\right)\ln C_{e}$$
where *q_e_* and *Ce* are, respectively, the adsorption amount and concentration of pollutants in the solution when the adsorption of contaminants by the adsorbent reaches equilibrium, mg·g^−1^ and mg·g^−1^, respectively; *q_m_* is the maximum adsorption capacity, mg·g^−1^; *K_L_*, *K_F_*, and *K_T_* are dimensional constants of Langmuir, Freundlich, and Temkin models, L·mg^−1^, L·mg^−1^, and L·g^−1^, respectively; 1/*n* is an experience constant with no gauge. *R* (8.314 × 10^−3^ kJ·mol^−1^·K^−1^) is the ideal gas constant, and *T* is the absolute thermodynamic temperature, *K*.

### 2.6. Adsorption Kinetics

Keeping the other conditions unchanged, the conical flask was placed in a constant-temperature shock chamber, and then the samples were taken at 5, 10, 15, 30, 60, 120, 180, 300, 480, 720, 1440, 2160, and 2880 min of the shock at a frequency of 150 r·min^−1^ without light. The supernatant was filtered by a 0.22 μm filter membrane, and ultraviolet spectrometry was used to determine the concentration of OTC solution; the experiment was repeated three times to average. The experimental results were fitted by the pseudo-first-order dynamics model [36], pseudo-second-order dynamics model [37], and intra-particle diffusion model [38,39,40].
$$\mathrm{pseudo}-\mathrm{first}-\mathrm{order}\;\mathrm{dynamics}\;\mathrm{model}:\;q_{t}=q_{e}\left(1-e^{-k_{1}t}\right)$$
$$\mathrm{pseudo}-\sec\mathrm{ond}-\mathrm{order}\;\mathrm{dynamics}\;\mathrm{model}:{\;q}_{t}=\frac{k_{2}q_{e}^{2}t}{1+k_{2}q_{e}t}$$
$$\mathrm{Intra}-\mathrm{particle}\;\mathrm{diffusion}\;\mathrm{model}:\;q_{t}=k_{3}t^{0.5}+C$$
where *t* is the adsorption time, min; *q_t_* and *q_e_* are the adsorption capacities at *t* and equilibrium, respectively, mg·g^−1^, mg·g^−1^; *K*_1_, *K*_2_, and *K*_3_ are, respectively, the pseudo-first-order, pseudo-second-order, and intra-particle diffusion dynamics rate constants, g·mg^−1^·h^−1^, g·mg^−1^·h^−1^, and g·mg^−1^·h^−0.5^. *C* is the adsorption constant of the intra-particle diffusion model.

### 2.7. Adsorption Thermodynamics

Thermodynamic analysis of adsorption can describe the driving forces and directions of the adsorption process [41]. Through the study of biochar adsorption pollutants at different temperatures, the change in thermodynamic parameters in the adsorption process was calculated, and the thermodynamic formulas were used to calculate the Δ*G*, Δ*H*, and Δ*S*. Its thermodynamic formula is as follows,
$$\begin{array}{c} \Delta G=-\mathrm{RT}\ln k_{d}\\ \Delta G=\Delta H-T\Delta S \end{array}$$
where Δ*G* is the change in Gibbs free energy, kJ·mol^−1^; Δ*H* is the enthalpy change, J·mol^−1^; Δ*S* is the entropy change, kJ·mol^−1^·K^−1^; *R* (8.314 × 10^−3^ kJ·mol^−1^·K^−1^) is the ideal gas constant, *T* is the absolute thermodynamic temperature, *K*; *k_d_* is the adsorption constant, and it comes from *q_e_*/*C_e_*. Δ*H* and Δ*S* are derived from the slope and intercept by mapping *T* by Δ*G*.

## 3. Results and Discussion

### 3.1. Materials Characterization

#### 3.1.1. Scanning Electron Microscopy (SEM) Analysis

The surface morphology and structure of BC, HBC, and nZVI-HBC were observed by SEM images (Figure 1). Among them, BC has a rich porous structure and uneven surface, with disordered pairs of pore structures and varying sizes. The pore structure of HBC is relatively regular and there are few impurities in the pores, which provide a lot of space for the attachment of nanometer iron. nZVI-HBC shows that the nanosized zero-valent iron particles are amorphous, and the surface is relatively rough. Due to magnetic influence, some particles are agglomerated together in a chain shape [42]. The results show that the nano-zero iron can adhere to the surface of biochar and effectively prevent the aggregation of nanoparticles.

#### 3.1.2. X-ray Diffraction (XRD) Analysis

XRD was performed on BC, HBC, and nZVI-HBC, and the results are shown in Figure 2. The XRD pattern of BC shows that it has sharp peaks and a crystal structure. The peak body is mainly due to the presence of SiO_2_ and CaCO_3_. The characteristic peak about 2θ = 25° of HBC is the characteristic diffraction peak of coconut shell carbonization 002. The characteristic peak at 2θ = 45° is the characteristic diffraction peak of coconut shell carbonization 100. For nZVI-HBC, it is evident that the XRD pattern shows the characteristic peak of zero-valent iron corresponding to 110 plane diffraction volume center cubes when the diffraction peak 2θ = 44.8°, indicating that nano-zero-iron successfully loaded on HBC by the liquid-phase reduction method [43]. Its peak shape shows a certain diffusion phenomenon, indicating that the nanoparticles are in an amorphous state. In addition, the characteristic peak of Fe_3_O_4_ corresponds to 2θ = 35.5°, meaning that a small part of surface nano-zero iron particles is oxidized to trivalent iron [27,44]. Surface oxidation produced by Fe_3_O_4_ largely prevents internal nZVI from contacting the air.

#### 3.1.3. Fourier-Transform Infrared Spectroscopy (FTIR) Analysis

In this study, the infrared spectrum of biochar at 4000–400 cm^−1^ was measured to analyze its surface functional groups. The comparison of the infrared spectra of BC, HBC, and nZVI-HBC biochar is shown in Figure 3. The FTIR spectra of BC and HBC are similar. The corresponding absorption peak at a wavelength about 3415 cm^−1^ is the vibration peak of -OH, the corresponding absorption peak at about 2356 cm^−1^ is the vibration peak of C≡N, the corresponding absorption peak at about 1634 cm^−1^ is of the C=O vibration peak, and the corresponding absorption peak at about 1067 cm^−1^ is the -OH vibration peak [45,46,47]. This may speculate that the surfaces of BC and HBC may contain amino and hydroxyl clumps. In the FTIR spectrum of nZVI-HBC, in addition to the absorption peaks that appear between the BC and HBC above, the wavelength of about 662 cm^−1^ corresponds to the absorption peaks of the Fe-O telescopic vibration [48,49], indicating the synthesis of nZVI, which is consistent with the previous XRD conclusions.

#### 3.1.4. X-ray Photoelectron Spectroscopy (XPS) Analysis

XPS was used to characterize the surface element morphology of nZVI-HBC. Figure 4 shows the XPS spectrum of nZVI-HBC, the spectrum of iron, and its relevant results. The spectrum of Fe 2p can determine the valence of iron elements on the surface of the material. As shown in the figure, the absorption peak at 705.97 eV is that of zero-valent iron, which proves that zero-valent iron successfully loaded on biochar by the liquid-phase reduction method [50]. The absorption peaks at 709.51 eV and 723.11 eV are characteristic peaks of Fe^2+^, and those at 711.73 eV and 725.33 eV are characteristic peaks of Fe^3+^ [51]. Among them, the appearance of Fe^2+^ and Fe^3+^ absorption peaks are caused by the oxidation of the Fe^0^ surface in contact with air to form different iron oxides.

### 3.2. Batch Adsorption Experiment

#### 3.2.1. Influence of Biochar Addition Amount on OTC Removal Effect

Figure 5 shows the effect of different biochar additions on the adsorption rate of OTC. Among them, the adsorption rate of OTC increases linearly with the increase in BC addition, and when the amount of BC addition is 20 mg, the adsorption rate of OTC is 79.925%. The removal rate of OTC increases rapidly when HBC addition is 2–10 mg, while the removal rate tends to be flat after 10 mg. For nZVI-HBC, when the dosage is 6 mg, the removal rate reaches 91.192%, and then the removal rate is not more than 7.49%. With the rise in the amount of nZVI-HBC added, the active sites involved in the reaction in the solution also increase, resulting in an increase in the removal rate. However, with the increasing amount of biochar added, there are too many available adsorption sites. Still, the concentration of antibiotics in the solution is limited, so the removal rate of antibiotics does not change [52]. To sum up, to obtain the best adsorption effect and economic benefit, the addition level of 6 mg was selected for subsequent experiments.

#### 3.2.2. Influence of Initial pH on OTC Removal Effect

Figure 6 shows the effect of biochar on the adsorption rate of OTC at different pH values. When pH = 3, BC and HBC have the highest adsorption rate, and OTC^+^ is the main form of OTC in the solution, which has electrostatic repulsion with positively charged biochar. At this time, the adsorption rates of BC and HBC are highest, indicating that, in addition to electrostatic interaction, there are mechanisms such as π–π EDA interaction during the adsorption of BC and HBC. With the increase in pH, the adsorption rate of the OTC decreases gradually and reaches the lowest value at pH = 11.

For nZVI-HBC, the removal rate of OTC is lowest when pH = 3, because under acidic conditions, iron begins to rapidly corrode to produce bivalent iron, which leads to the generation of large numbers of hydrogen and hydrogen free radicals. Moreover, reducing hydrogen may directly reduce and degrade OTC, resulting in the lowest OTC removal rate. When pH = 5, OTC has the highest removal rate, which is because the particles are positively charged, and the electrostatic repulsion between particles makes the particles more easily dispersed, thus providing more active sites and causing the adsorption of OTC [53].

In addition, the concentration of the OTC decreases slightly the alkaline environment. This is due to the high pH value, the OTC being a negative charge state, and the nZVI-HBC surface also being a negative charge state, resulting in static rejection between the material and OTC, and iron corrosion will form an oxidation layer on the surface of the material. The active sites on the material surface are reduced, leading to a gradual decrease in the removal rate.

#### 3.2.3. Influence of Cation on OTC Removal Effect

The effect of cation type on the adsorbent in aqueous solution is closely related to its valence, and the effect of common cations on biochar adsorption OTC was selected [54], such as Na^+^, K^+^, Mg^2+^, and Ca^2+^. The concentration of cations is 0.1 mol·L^−1^, and the OTC concentration is 20 mg·L^−1^, with the experimental volume of 50 mL. The inhibition of the adsorption rate of the four cations to OTC as shown in Figure 7 is as follows: Ca^2+^ >Mg^2+^ >K^+^ >Na^+^. Cations with relatively high valence state have a stronger inhibition on OTC removal efficiency. At the same time, it can also be seen that cations with larger ionic radius in the same valence state have stronger competitiveness, occupying more active sites on the surface of biochar, the OTC removal inhibition is more obvious.. Among them, the inhibition of cations to BC and HBC to remove OTC is not apparent. For nZVI-HBC, Mg^2+^ and Ca^2+^ may react with them to produce an iron oxide on the surface of the material to more inhibit the removal of pollutants.

#### 3.2.4. Influence of Anions on OTC Removal Effect

The anions of NO_3_^−^, CO_3_^2−^, and HCO_3_^−^ were selected, and their ionic concentration was 0.1 mol·L^−1^, experimented at an OTC concentration of 20 mg·L^−1^, and a volume of 50 mL. As shown in Figure 8, the three anions have different degrees of inhibition effect on OTC adsorption efficiency. Among them, for BC and HBC, the inhibition is CO_3_^2−^ > HCO_3_^−^ > NO_3_^−^, because the solution after hydrolysis of acid root ions is alkaline. At the same time, OTC molecules are in an anion state under alkaline conditions, thus inhibiting OTC removal.

For nZVI-HBC, the inhibition is HCO^3−^ > CO_3_^2−^ >NO_3_^−^, because have a higher redox potential of NO_3_^−^, nano-zero iron particles were given priority to reduce NO_3_^−^, and the formation of iron oxides leads to passivation on the surface of nZVI-HBC to reduce reaction activity, and reduces the reaction rate. CO_3_^2−^ will accelerate the corrosion of iron and promote the generation of bivalent iron. As the reaction proceeds, iron particles form sediments or complexes such as siderite, iron carbonate hydroxide, aragonite, or calcite and other passivation films, reducing the reactivity of zero-valent iron and reducing the removal of pollutants. HCO_3_^−^ may react with zero-valent iron to produce CO_3_^2−^, resulting in an initial pH increase in the solution, inhibiting the removal of OTC. In addition, the presence of HCO_3_^−^ may lead to the formation of Fe^2+^ after the corrosion of nZVI possibly reacting with HCO_3_^−^ to produce ferrous carbonate (FeCO_3_), which will inhibit the reactivity of metal materials.

#### 3.2.5. Influence of Ion Concentration on OTC Removal Effect

The research shows that electrolytes in the solution not only change the strength of the interaction between adsorbent and adsorbent in the solution due to electrostatic action, but also compete with the adsorption sites on adsorbent with antibiotics, thus affecting the adsorption removal efficiency. As shown in Figure 9, the introduction of Na^+^ ions inhibits the biochar adsorption of OTC to varying degrees. Still, with the increase in the concentration of Na^+^ ions, the removal rate does not change significantly. On the one hand, Na^+^ ions in solution can lead to an electrostatic effect, which affects the adsorption capacity of biochar to OTC. On the other hand, because of the solution with high salt ion concentration, OTC molecules are not easy to release from the whole, resulting in poor solubility, thereby reducing adsorption capacity. It is also possible that Na^+^ ions react with functional groups on the surface of biochar, or that it competes with the OTC molecules for the adsorption sites on the surface of biochar, thereby inhibiting the adsorption of the OTC [55].

#### 3.2.6. Adsorption Isotherm Analysis

The adsorption isotherm refers to the relationship between the equilibrium concentration (*C_e_*) of adsorption capacity (*q_e_*) when the adsorption reaches equilibrium under certain temperature conditions. Figure 10 shows the adsorption effects of biochar to different concentrations of OTC at ambient temperatures of 288.15 K, 298.15 K, and 308.15 K. The fitting data are shown in Table 1. In general, with the increase in temperature, the adsorption capacity of BC, HBC, and nZVI-HBC to OTC gradually increases, the initial concentration of OTC increases, and the adsorption of biochar also steadily increases. Among them, the adsorption of OTC by BC, HBC, and nZVI-HBC conforms to the Langmuir model. The linear correlation coefficients R^2^ are better than 0.9848, which is much larger than the linear correlation coefficient of the Freundlich model (R^2^ > 0.8712). In the Temkin model, the linear correlation coefficients R^2^ are greater than 0.9845.

In the Langmuir model, *q_m_* and *K_L_* increase with ambient temperature, indicating that the higher the temperature, the higher the adsorption site of BC, HBC, and nZVI-HBC materials and OTC molecular adsorption, with good adsorption effect, and the adsorption process is heat absorption. In the Freundlich model, *n* > 1 and increases with temperature, indicating that OTC is readily adsorbed to the biochar surface. As the ambient temperature increases, the value of *K_F_* increases, indicating that the adsorption capacity of the material also increases. In the Temkin model, the “b” value decreases with the temperature, implying that the adsorption affinity between the adsorption active sites and OTC molecules is also increasing.

According to the simulation results, when the ambient temperature is 308.15 K, the best adsorption capacity of BC, HBC, and nZVI-HBC to OTC is 78.6155 mg·g^−1^, 130.6579 mg·g^−1^, and 196.6985 mg·g^−1^.

#### 3.2.7. Adsorption Thermodynamics

To reveal the adsorption behavior of BC, HBC, and nZVI-HBC, the thermodynamics were calculated. The thermodynamic parameters of BC, HBC, and nZVI-HBC biochar for OTC in solution are shown in the table.

As shown in Table 2, the ambient temperature is 288.15 K, 298.15 K, and 308.15 K, and BC, HBC, and nZVI-HBC all have negative values for OTC adsorption of ΔG, indicating that the adsorption reaction is spontaneous and consistent with previous research results. |ΔG| increases with the ambient temperature, suggesting that with the rise in ambient temperature, the degree of the spontaneity of the reaction increases. The value of ΔH is positive, meaning that the adsorption process is endothermic, and the higher the temperature, the better the degree of adsorption. The values of ΔS are positive, indicating that the reaction is an entropy increase process, the adsorption reversibility is poor, and the randomness of the solid–liquid interface increases with the temperature [56].

#### 3.2.8. Adsorption Kinetics

Figure 11 shows the effect of adsorption time on the removal of OTC by BC, HBC, and nZVI-HBC. The results of the adsorption dynamics fit are shown in Table 3. As shown in the table, the correlation coefficients in the pseudo-first-order and pseudo-second-order dynamics model are above 0.9064, the BC, HBC, and nZVI-HBC are more in line with the pseudo-second-order dynamics model, and the linear correlation coefficients R^2^ are 0.9760, 0.9723, and 0.9976, respectively, which better describes the adsorption dynamics of OTC on adsorbents. In addition, the adsorption capacity (*q_e_*) of BC, HBC, and nZVI-HBC to OTC at equilibrium is closer to the theoretical value derived from the pseudo-second-order dynamics model.

The results show that the adsorption of BC, HBC, and nZVI-HBC on OTC is mainly based on chemical adsorption. The adsorption can quickly achieve adsorption balance, which is related to the OTC molecular structure containing more aromatic ring structures. The three biochars can be reacted with OTC through the π–π interaction [57].

To better explain the adsorption mechanism, it was further fitted by the intra-particle diffusion model, and the results show that OTC adsorption on biochar is divided into three stages. The fitting results are shown in Table 4. The first stage is rapid film diffusion, and due to the presence of a large number of adsorption sites, the reaction takes place quickly. The second stage is the slow particle internal adsorption stage. Here, the adsorption site of the biochar surface is gradually saturated, and the adsorption rate gradually decreases. The third stage is the adsorption-desorption balance stage. Here, adsorption has reached a balance, the diffusion rate in the pore decreases, and the adsorption will not increase with time. The linear correlation coefficient R^2^ of each part is also more than 0.9176, and its intercept “C” gradually increases with time, indicating that the film diffusion has a more substantial effect.

### 3.3. Analysis of Degradation Products and Adsorption Mechanism

#### 3.3.1. Degradation Products

During the reaction of nZVI-HBC to remove OTC, the metabolites produced in the reaction process were detected and analyzed by mass spectrometry. Its degradation products were mainly determined by different mass ratios in mass spectrometry [58,59]. The highest peak in m/z 461 is the ion peak of OTC, which proves that the primary substance in the original solution is OTC.

With the addition of nZVI-HBC, the ion peak strength of OTC is reduced, meaning that the concentration of OTC decreases, as well as the emergence of ion peaks of other strengths, indicating that OTC degrades during removal and produces new substances. The degradation path and degradation product of OTC are inferred from mass spectrometry; the results shown in Figure 12.

As shown in Figure 12, there are two main degradation paths. One path is: the generation of “B” is due to the loss of two N-methyl groups over the OTC. “C” is then formed by the removal of the amide group, based on which the absence of hydroxyl and amino groups is converted into “D,” and “E” is generated through ring-opening reactions, dihydroxylation reactions, and the removal of an acetyl group. The formation of “F” is due to the ring-opening reaction of “E,” the removal of a propyl group, and the removal of carboxyl, with an hydroxyl group replacing an acetyl group. The absence of methyl, carboxyl, and aldehyde groups on “F” and the ring-opening reaction produce “G.”

The second path is: the loss of methyl, carboxyl, and amide groups on “A” forms “H,” while removing methyl and amino groups on “H” generates “I.” “J” is formed by the removal of an hydroxyl group from “I.” “J” is then converted into “K” by a ring-opening reaction, with the reduction of ethyl, addition reaction, and demethylation. The removal of the aldehyde group and hydroxyl group on “K” produces “L,” in which “L” can form “M” through demethylation and the removal of the hydroxyl group. “L” can also be further developed by ring-opening reactions, the removal of propyl groups, and hydroxyl substitution of acetyl groups to form “N”. The formation of “O” is due to the removal of hydroxyl groups from “N.” “P” is formed on “O” by demethylation, ring-opening, and elimination reactions.

#### 3.3.2. Mechanism Analysis

By LC-MS analyses, the removal of OTC by material nZVI-HBC involves the following aspects:(1)nZVI-HBC, because of its biochar adsorption properties and adsorption of OTC to the surface of the material, enhances the contact between pollutants and biochar;(2)Part of the OTC is adsorbed and fixed to the surface by nZVI-HBC, and with the zero-valent iron reaction, partial degradation occurs;(3)The main degradation reactions are oxidation reaction, ring-opening reaction, and the removal of functional groups;(4)Some of the Fe^2+^ and Fe^3+^ products generated by the zero-valent iron in the air also adsorb a certain amount of OTC.

In summary, nZVI-HBC to OTC removal mainly includes adsorption and degradation.

## 4. Conclusions

(1) In this paper, the coconut shell was burned, and then the hydrochloric acid impregnation method was used, and the liquid-phase reduction method successfully prepared nZVI-HBC, and a series of characterization analyses was carried out to prove that nZVI was successfully loaded on HBC. In addition, nZVI can be well dispersed on the surface of HBC, reducing the agglomeration of zero-valent iron.

(2) BC, HBC, and nZVI-HBC had specific removal effects on OTC, of which the removal effects were nZVI-HBC > HBC > BC. The adsorption capacity of nZVI-HBC to OTC was up to 196.70 mg·g^−1^. The experiment showed that increasing the amount of biochar added was beneficial to OTC removal, and it was more beneficial to OTC removal under acidic conditions. It was shown that nZVI-HBC was a material with good adsorption performance and potential utilization value.

(3) The removal of OTC by nZVI-HBC included both adsorption and degradation. nZVI played an essential role in the removal of OTC. The degradation products and degradation path of OTC were inferred by LC-MS. Through the above experimental analysis, OTC was adsorbed on the material surface by nZVI-HBC, because of nZVI reduction of the surface adsorption of pollutants degradation removal.

## Figures and Tables

**Figure 1 ijerph-18-13107-f001:**
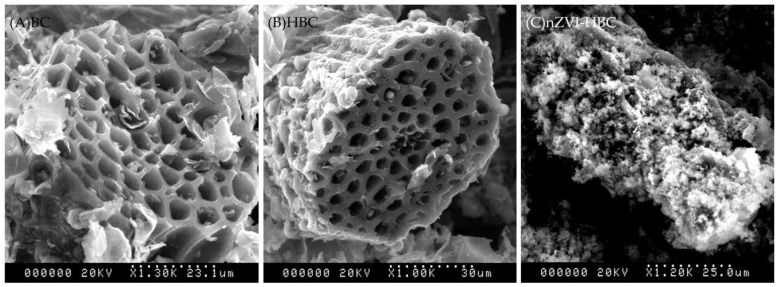
SEM images of biochar. (**A**) BC, (**B**) HBC, and (**C**) nZVI-HBC.

**Figure 2 ijerph-18-13107-f002:**
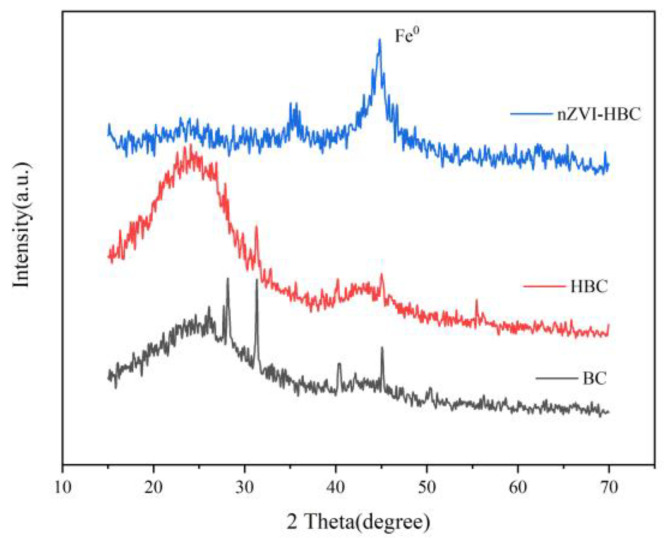
X-ray diffraction patterns of biochar.

**Figure 3 ijerph-18-13107-f003:**
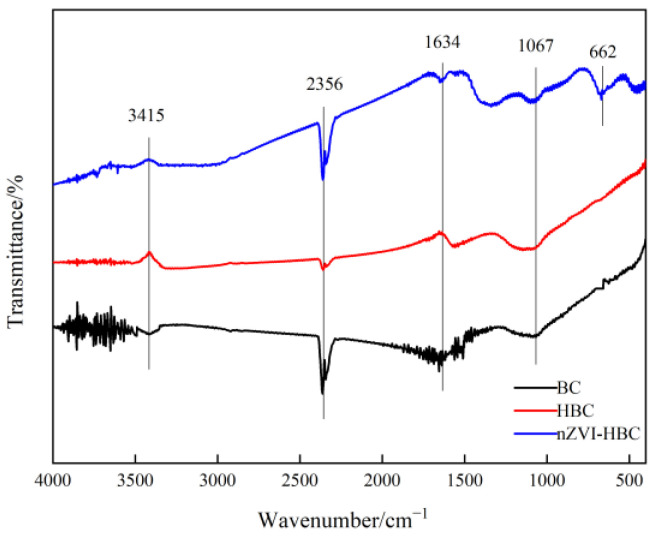
Fourier-transform infrared spectroscopy of biochar.

**Figure 4 ijerph-18-13107-f004:**
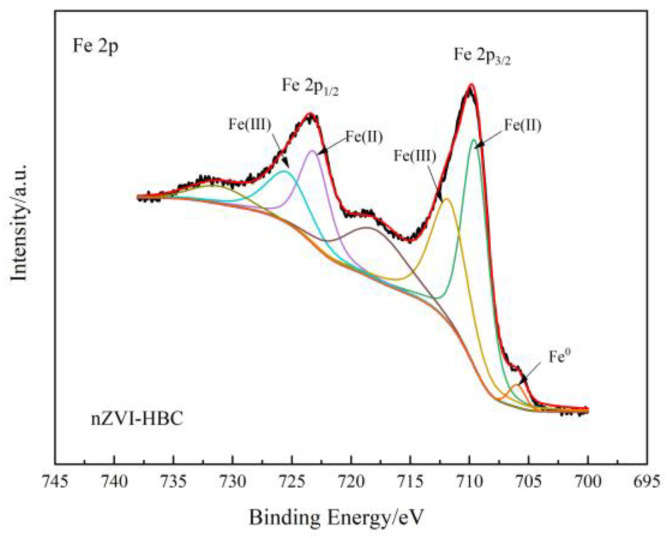
XPS diagram of nZVI-HBC.

**Figure 5 ijerph-18-13107-f005:**
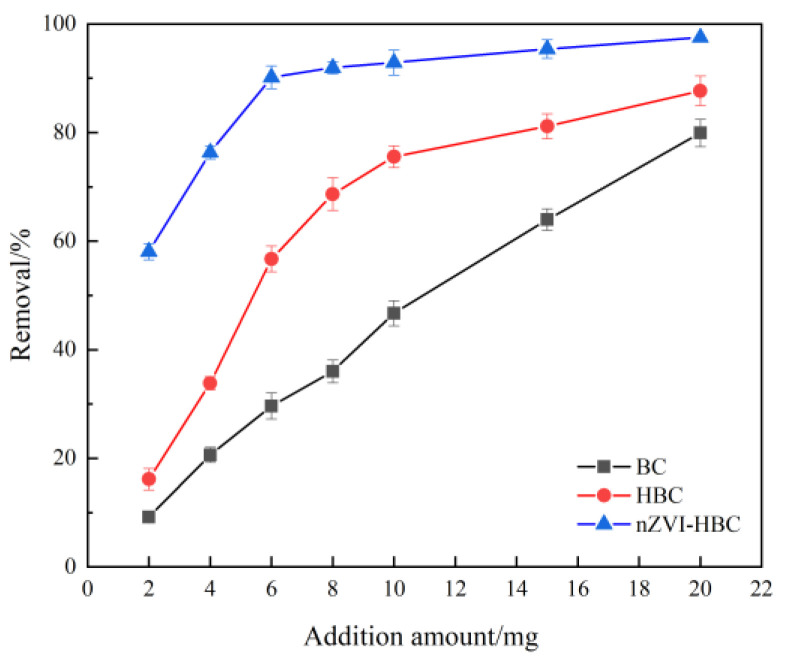
The effect of biochar addition on OTC removal rate.

**Figure 6 ijerph-18-13107-f006:**
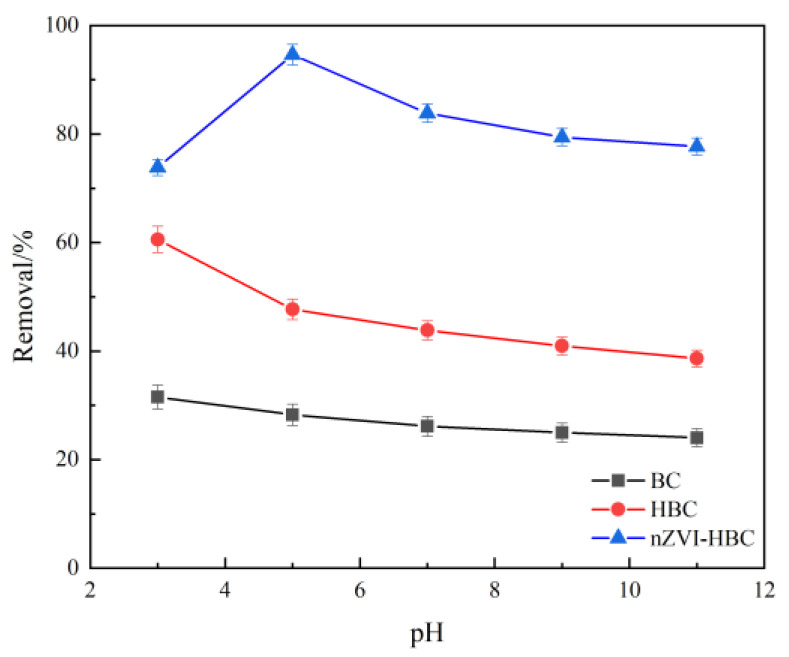
The effect of biochar addition on OTC removal rate.

**Figure 7 ijerph-18-13107-f007:**
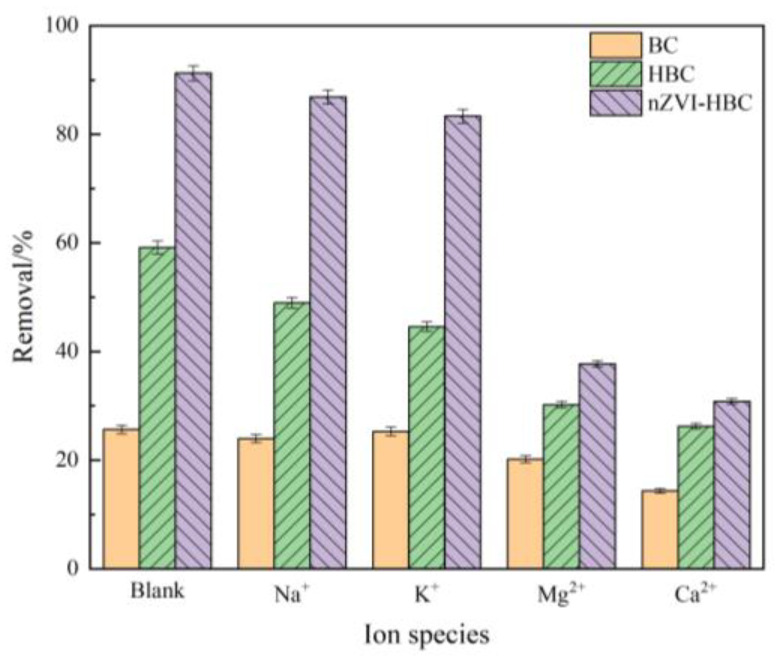
The effect of cations on OTC removal rate.

**Figure 8 ijerph-18-13107-f008:**
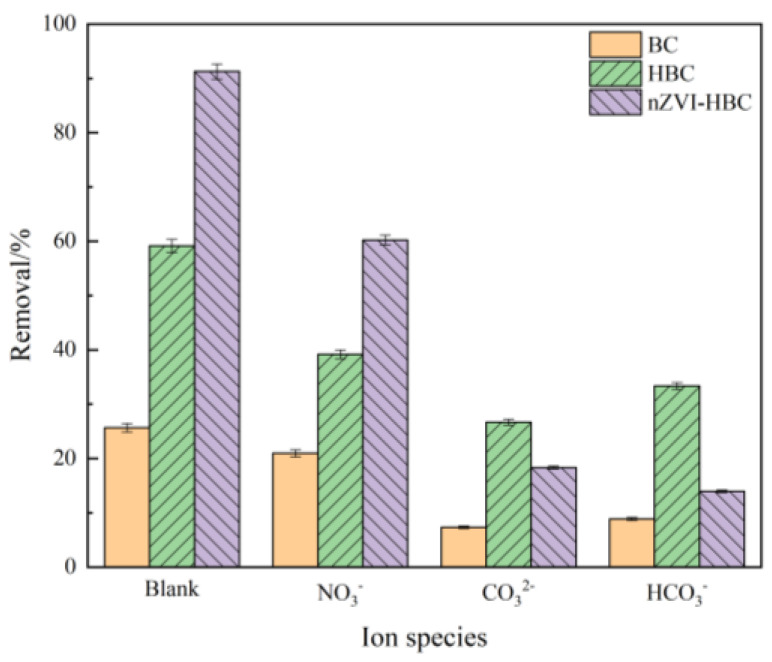
The effect of anions on OTC removal rate.

**Figure 9 ijerph-18-13107-f009:**
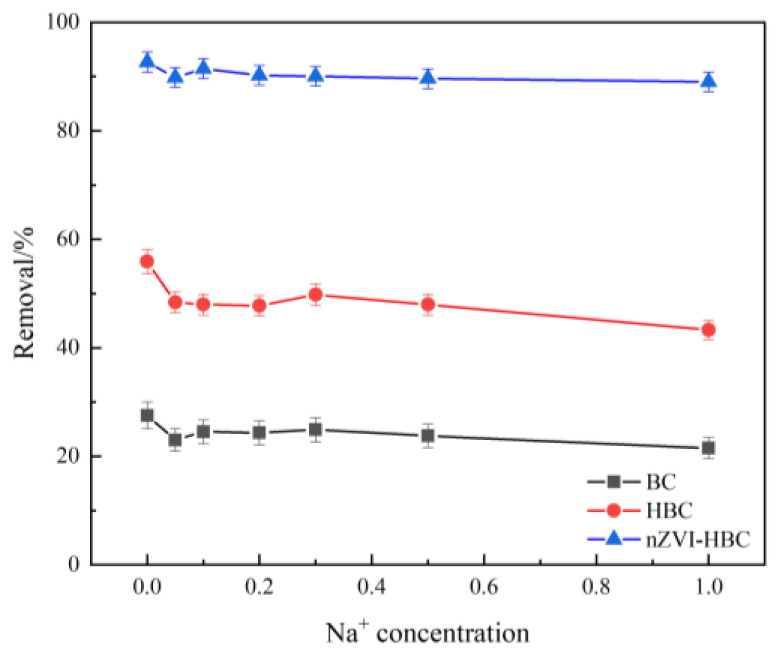
The effect of Na^+^ concentration on OTC removal rate.

**Figure 10 ijerph-18-13107-f010:**
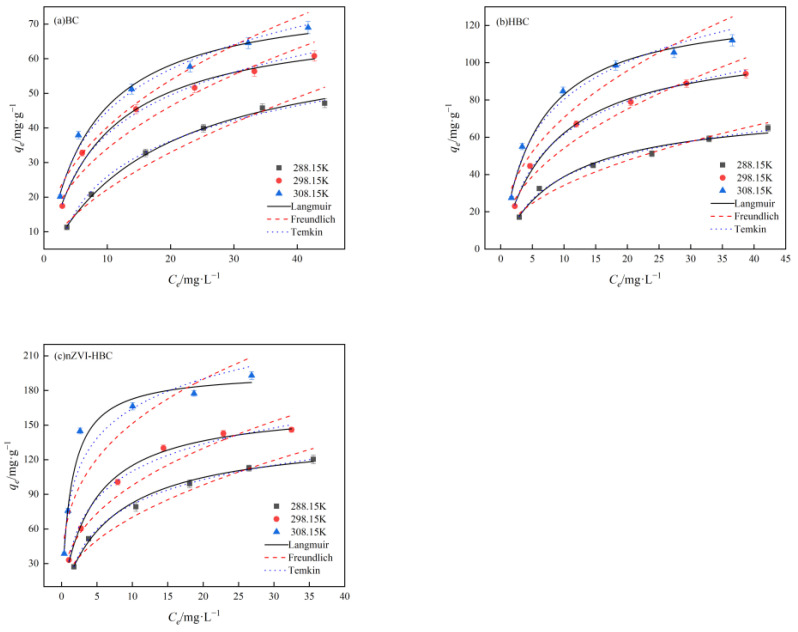
Adsorption isotherms of OTC with (**a**) BC, (**b**) HBC, and (**c**) nZVI-HBC.

**Figure 11 ijerph-18-13107-f011:**
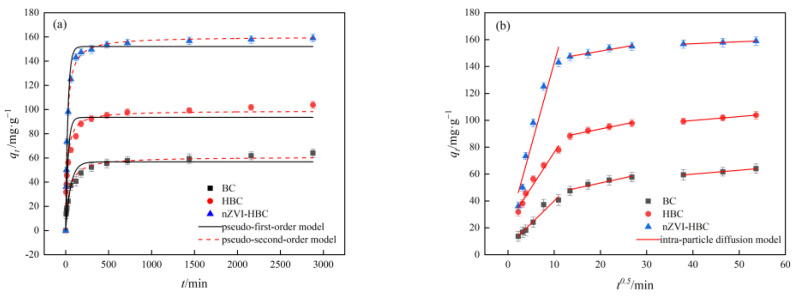
Adsorption kinetics curves of OTC adsorption of BC, HBC, and nZVI-HBC. (**a**) pseudo-first-order and pseudo-second-order dynamics model. (**b**) intra-particle diffusion model.

**Figure 12 ijerph-18-13107-f012:**
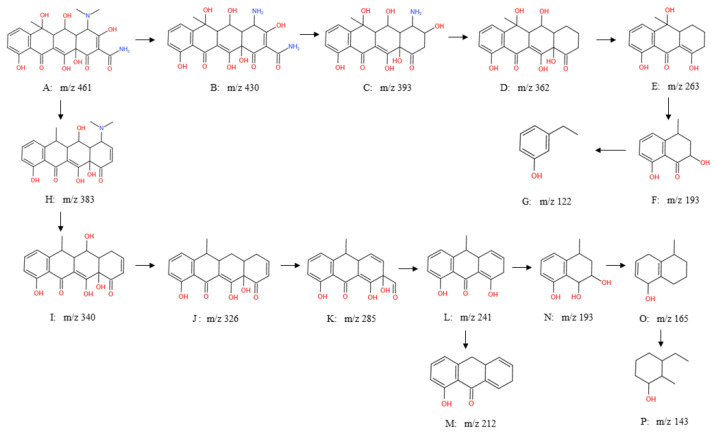
Degradation pathway of OTC.

**Table 1 ijerph-18-13107-t001:** Adsorption isothermal equation fitting parameters of OTC adsorption of BC, HBC, and nZVI-HBC.

	*T*/*K*	Langmuir			Freundlich			Temkin		
		*q_m_*/mg·g^−1^	*K_L_*/L·mg^−1^	*R* ^2^	*n*	*K_F_*/mg·L^−1^	*R* ^2^	*B*	*K_T_*/L·mg^−1^	*R* ^2^
	288.15	67.6953	0.0568	0.9977	1.7603	6.0044	0.9681	0.1623	0.5816	0.9974
BC	298.15	71.9277	0.1174	0.9892	2.2525	12.2370	0.9405	0.1520	1.0383	0.9905
	308.15	78.6155	0.1421	0.9868	2.3783	15.2701	0.9416	0.1457	1.2748	0.9877
	288.15	76.7222	0.1025	0.9858	2.0997	11.4201	0.9555	0.1379	0.9300	0.9924
HBC	298.15	113.3720	0.1217	0.9927	2.1155	18.2221	0.9465	0.0975	1.1297	0.9967
	308.15	130.6579	0.1735	0.9855	2.1560	25.7669	0.9131	0.0882	1.5901	0.9845
	288.15	142.7134	0.1372	0.9968	2.0598	22.9187	0.9726	0.0773	1.3756	0.9992
nZVI-HBC	298.15	167.2904	0.2199	0.9957	2.4370	37.9853	0.9582	0.0723	2.4721	0.9979
	308.15	196.6985	0.7208	0.9848	3.0709	71.5141	0.8712	0.6895	8.3281	0.9956

**Table 2 ijerph-18-13107-t002:** Thermodynamic parameters of OTC adsorption of BC, HBC, and nZVI-HBC.

	*T*/*K*	Δ*G*/KJ·mol^−1^	Δ*H*/KJ·mol^−1^	Δ*S*/KJ·mol^−1·^K^−1^
	288.15	−7.8202		
BC	298.15	−9.8900	32.7392	0.1409
	308.15	−10.7110		
	288.15	−9.2327		
HBC	298.15	−9.9790	17.6457	0.0929
	308.15	−11.2227		
	288.15	−9.9326		
nZVI-HBC	298.15	−11.4459	61.5413	0.2469
	308.15	−14.8714		

**Table 3 ijerph-18-13107-t003:** Pseudo-first-order and pseudo-second-order dynamics model fitting parameters of OTC adsorption of BC, HBC, and nZVI-HBC.

	Pseudo-First-Order Dynamics Model			Pseudo-Second-Order Dynamics Model		
	*k_1_*/min^−1^	*q_e_*/ mg·g^−1^	*R* ^2^	*k_2_*/min^−1^	*q_e_*/ mg·g^−1^	*R* ^2^
BC	0.0174	56.7510	0.9248	0.0004	60.8867	0.9760
HBC	0.0361	93.5487	0.9064	0.0005	99.0229	0.9723
nZVI-HBC	0.0376	152.1172	0.9854	0.0003	160.2055	0.9976

**Table 4 ijerph-18-13107-t004:** Intra-particle diffusion model fitting parameters of OTC adsorption of BC, HBC, and nZVI-HBC.

	*K*_i1_ /mg·g^−1^·min^−0.5^	*C* _1_	*R* _1_ ^2^	*K*_i2_ /mg·g^−1^·min^−0.5^	*C* _2_	*R* _2_ ^2^	*K*_i3_ /mg·g^−1^·min^−0.5^	*C* _3_	*R* _3_ ^2^
BC	3.3681	6.3466	0.9379	0.7462	38.4293	0.9281	0.2888	48.4239	0.9952
HBC	5.2774	23.2910	0.9596	0.7041	79.4408	0.9543	0.2898	88.2895	0.9990
nZVI-HBC	12.4327	18.3370	0.9176	0.5996	139.4657	0.9457	0.1444	151.2029	0.9952

## Data Availability

The study did not report any data.

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
