# Peer review of "Mechanism of Oxytetracycline Removal by Coconut Shell Biochar Loaded with Nano-Zero-Valent Iron"

_ijerph, 2021, doi:10.3390/ijerph182413107_

Round 1

Reviewer 1 Report

Comments regarding manuscript “Mechanism of oxytetracycline removal by coconut shell biochar loaded with Nano zero-valent iron”

Generally, the manuscript is well-written and comprehensive, however I have some comments.

Introduction – please indicate the novelty of the study. How important is to remove the OTC from water right now? What is the concentration of OTC in water or wastewater?

Abstracts – full stops not semi commons – please change in the first few sentences.

Abbreviations – please check carefully all the text of manuscript. Sometime you wrote both full name and abbreviation too. Please change it.

The Authors use capital letters in the middle of sentences (l. 35; l. 56) Why? . Please check carefully all the manuscript and change.

Paragraph 2.2: What is it? It not methodology, just listed some phrases... Provide description of point 2.2!

2.1 What is grain size of the material? Add information.

  1. 101 What does it mean “operated under nitrogen protection”?

Pseudo-first/second kinetic model – please change. Provide references to all equitation.

Fig 1. Add A, B, C at the photos.

At the moment, there is no discussion of provided results. Please add information about previous used the coconut shell biochar as a sorbent. Add paragraph about other ways of removal the OTC. What is practical implementation of your research? Please add information.

Author Response

Thank you for your careful comments and suggestions. The answers below are our revisions and some thoughts based on your suggestions. The detailed revisions are marked in red in the revised manuscript. The following is a point-to-point response.

Reviewer 2 Report

I recommend the following minor adjustments

  • Line 78: I recommend deleting chapter 2.2 Experimental equipment and analytical instruments, which only lists the instruments used. Indicate the instrument used with the appropriate method, as usual. Some devices lack information about the device manufacturer and country of origin (muffle furnace, etc.).
  • Line 112: Chapter 2.5. Batch adsorption experiment, namely experiment 2 Effect of solution pH on OTC removal rate. How the pH was controlled throughout the experiment (24 h). Has the pH value changed? Has a constant pH been ensured at all times? Please complete.
  • Line 121 add a space between the unit and the number: 50mL.
  • Line 148 space after the semicolon: L · g1; 1 / n.
  • Line 148 and 174  - please repair the molar gas constant unit.
  • Line 149 and 175 - absolute temperature scales are Kelvin, K (capital letter).
  • Line 226, 228, 229 - Fill in the space between the value and the unit (eV).
  • Line 158-161 - parameter markings must be identical. There is qe in the formula, Qe in the explanatory notes, etc.
  • Line 348 - tab. 1 and line 391 - tab. 3 replenish units.

Author Response

Thank you for your careful comments and suggestions. The following answers are our revision and some thoughts according to your suggestions. Detailed revised portion are marked in red in the revised manuscript. The following are point-to-point responses. 

Round 2

Reviewer 1 Report

I have no more comments.